# Analysis of 18 Free Amino Acids in Honeybee and Bumblebee Honey from Eastern and Northern Europe and Central Asia Using HPLC-ESI-TQ-MS/MS Approach Bypassing Derivatization Step

**DOI:** 10.3390/foods11182744

**Published:** 2022-09-07

**Authors:** Fredijs Dimins, Ingmars Cinkmanis, Vitalijs Radenkovs, Ingrida Augspole, Anda Valdovska

**Affiliations:** 1Faculty of Food Technology, Latvia University of Life Sciences and Technologies, LV-3004 Jelgava, Latvia; 2Division of Smart Technologies, Research Laboratory of Biotechnology, Latvia University of Life Sciences and Technologies, LV-3004 Jelgava, Latvia; 3Processing and Biochemistry Department, Institute of Horticulture, LV-3701 Dobele, Latvia; 4Faculty of Agriculture, Latvia University of Life Sciences and Technologies, LV-3001 Jelgava, Latvia; 5Division of Agronomic Analysis, Research Laboratory of Biotechnology, Latvia University of Life Sciences and Technologies, LV-3002 Jelgava, Latvia

**Keywords:** bee honey, bumblebee, essential amino acids, food composition, LC-MS, sugars

## Abstract

The profile of amino acids and mono- and disaccharides in conventional polyfloral honey originated from Latvia and Tajikistan and less found in nature bumblebee honey from Russia was investigated. The analysis of free amino acids (FAAs) accomplished by multiple reaction monitoring (MRM) using triple quadrupole mass selective detection (HPLC-ESI-TQ-MS/MS) revealed the presence of 17 FAAs. The concentration of FAAs varied in the range of 0.02–44.41 mg 100 g^−1^ FW. Proline was the main representative of FAAs, contributing to the total amount of FAAs from 41.7% to 80.52%. The highest concentration of proline was found in bumblebee and buckwheat honey, corresponding to 44.41 and 41.02 mg 100 g^−1^, respectively. The concentration of essential amino acids (AAs), i.e., leucine, and isoleucine was found to be the highest in buckwheat honey contributing up to 12.5% to the total amount of FAAs. While, the concentration of branched-chain AAs fluctuated within the range of 1.08–31.13 mg 100 g^−1^ FW, with buckwheat honey having the highest content and polyfloral honey the lowest, respectively. The results of this study confirmed the abundance of FAAs both in honeybee and bumblebee honey. However, the concentration of individual FAAs, such as proline, aspartic acid, leucine, and isoleucine in bumblebee honey was many folds higher than observed in honeybee polyfloral honey.

## 1. Introduction

Historically honey originating from bees (*Apis* genus) was used in medicine for treating various ailments, incl., intestinal disorders, cancer, asthma, tuberculosis, thirst, hiccups, fatigue, dizziness, hepatitis, constipation, etc. [1]. Due to its unique antimicrobial properties, the application of honey was found to be effective in treating infected wounds. Its superior inhibitory activity against pathogenic microorganisms has repeatedly been proven [2,3]. However, less explored is “bumblebee honey” or “bumblebee mead” which is produced by bumblebee (Bombus genus) and recognized by traditional medicine as it contains fewer toxic compounds than that bee honey [4,5]. According to a comprehensive article made available by Svangberg and Berggren [6] becomes clear that various regions in Germany and France from ancient times used bumblebee honey in folk. Besides, old Moravian people believed that if honey was collected from bumblebees in the field while maintaining silence, after bringing it to the altar parishioner will become rich. Even though honey was used by Ancient Egyptians, Assyrians, Chinese, Greeks, and Romans, and its multiple health-promoting properties were further supported by the scientific community, its uses in modern medicine are still way too limited [7]. Following the report of Madebekin [8], bumblebee honey has a much richer nutritional composition and biochemical profile than bee honey. It has been highlighted that 100 g of bumblebee honey could deliver as many enzymes, microelements, and vitamins as 1 kg of bee honey would. Superior nutritional composition could be explained by the way the honey is produced by bumblebees. It is known that bumblebees make honey from collected pollen and nectar by chewing and mixing it with their saliva for a longer time. Besides, bumblebees, unlike bees, do not accumulate honey for the winter so at the time of harvesting it is relatively fresher than bee honey. The produced honey is enough till the time when the queen hibernates, spending the winter safely underground.

One of the nutritional characteristics of the food is the content of AAs and proteins since they are involved in many vital processes such as the synthesis of proteins, hormones, and neurotransmitters. The availability of essential AAs in honey makes this product unique from a nutritional standpoint of view [9]. However, the honey evaluation criteria discussed in the Council Directive 2001/110/EC [10] and amended Codex Alimentarius Standard for Honey [11] do envisage standardized protocols for the determination of basic parameters such as the content of moisture, reducing sugars, and hydroxymethylfurfural, free acidity, diastase activity, and electrical conductivity. Quantitative determination of AAs, however, due to the complexities both in sample preparation and analysis is ignored [12].

With the advancement in analytical chemistry and instrumentation, the analysis of FAAs becomes more easily done by bypassing derivatization steps and without taking a risk of environmental pollution and human health matter. The advantages of liquid chromatography-mass spectrometry (LC-MS) in the analysis of FAAs in various foodstuff, including barley extracts [13], orange juice [14], mammalian milk [15], and honey [16,17] has repeatedly been confirmed, highlighting also robustness, sensitivity, and selectivity of this approach compared with HPLC-DAD [18] or GC-MS [19] techniques. Selective analysis of FAAs utilizing the LC-MS system makes it attainable to distinguish both the botanical origin of the honey and verify its authenticity [17] in a relatively shorter time. This statement was reinforced by dos Santos Scholz et al. [20], indicating that some individual FAAs, e.g., proline, which is the most abundant in original honey, can be used as indicators of the geographical origin.

The limited information on the profile and composition of FAAs in the honey of various origins, bumblebee and buckwheat in particular prompted the design of this study focusing on the evaluation of FAAs and individual sugars in five honey of Eastern and Northern Europe and Central Asia.

## 2. Materials and Methods

### 2.1. Honey Samples

Honey samples for the research were randomly collected in three years of study: 2019, 2020, and 2021 originating from Eastern and Northern Europe and Central Asia. Five honey samples in total: 3 polyfloral, 1 buckwheat flower, and 1 polyfloral (forest) were analyzed considering the composition and profile of FAAs. A detailed information on collected honey samples is given in Table 1.

### 2.2. Chemicals and Reagents

Commercial standards, i.e., a mixture of 18 AAs, xylose (Xyl), arabinose (Ara), fructose (Fru), glucose (Glu), sucrose (Suc), maltose (Mal), and glycerol (Gly) were purchased from Sigma-Aldrich Chemie Ltd., (Steinheim, Germany). Acetonitrile (MeCN) and formic acid (HCOOH) (puriss r.a.) of liquid chromatography-mass spectrometry (LC-MS) grade were purchased from the same producer. The cation-exchange resin “DIAION™ UBK550” was provided in kind by the company Mitsubishi Chemical Corporation (Tokyo, Japan) for laboratory purposes. Ammonium hydroxide solution (NH_4_OH) (25% *v*/*v*) was obtained from Chempur (Piekary Śląskie, Silesia, Poland). Hydrochloric acid (HCl) (37% *v*/*v*) was purchased from VWR™ International, GmbH (Darmstadt, Germany). Ultrapure water was produced using the reverse osmosis “PureLab Flex Elga” water purification system (Veolia Water Technologies, Paris, France). Buffer solutions used for cleaning up and desorption of AA from the cation-exchange resin were as follows:buffer A 10 mM hydrochloric acid solution (pH 2.04);buffer B 6 M ammonium hydroxide solution (pH 11.14).

### 2.3. Preparation of Amino Acids for HPLC-ESI-TQ-MS/MS Analysis

Isolation and purification of AA from honey matrix were performed based on the protocol described by Cukier et al. [21]. Briefly, 1.0 g honey with accuracy ± 0.01 g was transferred into 15.0 mL conical centrifuge tubes (Sarstedt AG & Co. KG, Nümbrecht, Germany), and 10.0 mL of H_2_O was added. Afterward, the mixture was subjected to 1 min intensive Vortexing using the “ZX3” vortex mixer (Velp^®^ Scientifica, Usmate Velate, Italy), followed by centrifugation at 10,280× *g* for 10 min at 4 ± 1 °C in a “Hermle Z 36 HK” centrifuge (Hermle Labortechnik, GmbH, Wehingen, Germany). Filtration of collected supernatant was done using a 0.22 µm polyvinylidene fluoride (PVDF) hydrophilic membrane filter (Durapore, Millipore, Billerica, MA, USA). Conditioning/equilibration of cation-exchange “DIAION™ UBK550” fractionation resin (90.0 mg in 3 mL gravity column) was performed using 3 mL of buffer A solution at a rate of 1 mL min^−1^ under the pressure −2.0 ± 0.1 ″Hg ensured by 12 port “Chromabond SPE” system (Macherey-Nagel GmbH & Co. KG, Dueren, Germany) coupled with N 820 LABOPORT vacuum pump (KNF Neuberger AB, Stockholm, Sweden). The loaded sample was washed with 6 mL H_2_O and a flow-through fraction was collected for estimation of the presence of AA. For desorbing AAs from the stationary phase of cation-exchange resin, a 3.0 mL buffer B solution was used. The collected eluate fraction was subjected to drying using a “Laborota 4002” rotary evaporator (Heidolph, Swabia, Germany) at 40 ± 1.0 °C and 60.0 ± 2.0 mBar. The obtained dry residues were kept at a temperature of −18.0 ± 1.0 °C until further analysis and use, a maximum of 72 h. A schematic representation of sample preparation steps is depicted in Figure 1.

### 2.4. The HPLC-ESI-TQ-MS/MS Analytical Conditions for Amino Acids

The chromatography analysis of AA (Figure 2) was performed on a “Shimadzu Nexera UC” series liquid chromatography (LC) system (Shimadzu Corporation, Tokyo, Japan) coupled to a triple quadrupole mass-selective detector (TQ-MS-8050, Shimadzu Corporation, Tokyo, Japan) with an electrospray ionization interface (ESI). A sample of 3 µL was injected onto a reversed-phase “Discovery^®^ HS F5-3” column (3.0 µm, 150 × 2.1 mm, Merck KGaA, Darmstadt, Germany) operating at 40 °C and a flow rate of 0.25 mL min^−1^. The mobile phases used were acidified H_2_O (1.0% HCOOH *v*/*v*) (A) and acidified MeCN (1.0% HCOOH *v*/*v*) (B). The program of stepwise gradient elution of the mobile phase B for 20 min was implemented as follows: T_0_ min = 5.0%, T_5_._0_ min = 30.0%, T_11_._0_ min = 60.0%, T_12_._0_ min = 80.0%, T_12_._1_ min = 5.0%. Finally, re-equilibration for 3 min was done after each analysis following the conditions of the initial gradient. The MeCN injections were included after each sample as a blank run to avoid the carry-over effect. Data were acquired using “LabSolutions Insight LC-MS” version 3.7 SP3, which was also used for instrument control and processing. The ionization in positive ion polarity mode was applied in this study, while data were collected in profile and centroid modes, with a data storage threshold of 5000 absorbance for MS. The operating conditions were as follows: detector voltage 1.98 kV, conversion dynode voltage 10.0 kV, interface voltage 4.0 kV, interface temperature 300 °C, desolvation line temperature 250 °C, heat block temperature 400 °C, nebulizing gas argon (Ar, purity 99.9%,) at a flow rate of 3.0 L min^−1^, heating gas carbon dioxide (CO_2_, purity 99.0%) at a low of 10.0 L min^−1^, and drying gas nitrogen (N_2_, separated from air using a nitrogen generator system from “Peak Scientific Instruments Ltd.” (Inchinnan, Scotland, UK), purity 99.0%) at flow 10.0 L min^−1^. All AAs were observed in the programmed and optimized multiple reaction monitoring (MRM) mode.

### 2.5. Preparation of Standard Stock Solution

Stock solution containing 0.025 uM mL^−1^ AAs was prepared in 10.0 mL 20% acidified MeCN solution (MeCN:H_2_O:formic acid ratio 80:19:1 *v*/*v*/*v*). Quantification of AAs was done by injecting 3.0 μL at 15 °C of calibration solution with the range of 0.0025–0.20 μM L^−1^. The working solution was prepared immediately before being used. Representative chromatographic separation of 18 AAs is given in Figure 2.

### 2.6. The HPLC-RID Conditions for Carbohydrates Analysis

Quantitative analysis of mono- and disaccharides (Figure 3) in honey was accomplished on a “Waters Alliance” HPLC system (model No. e2695) coupled to a “2414 RI” detector and a “2998 column heater” (Waters Corporation, Milford, MA, USA) following the methodology described by Radenkovs et al. [22]. Chromatographic separation was done on an Altima Amino (4.6 × 250 mm; 5 μm; Grace™, Columbia, MD, USA) column. The column and flow cell temperature was maintained at 30 °C. A mixture of H_2_O and MeCN (80:20, *v*/*v*) was used as the mobile phase in isocratic mode. The flow rate of the mobile phase was 1.0 mL min^−1^. The injection volume was 15 μL. System control, data acquisition, analysis, and processing were performed using Empower 3 Chromatography Data Software version (build 3471) (Waters Corporation, Milford, MA, USA).

### 2.7. Statistical Analysis

The results obtained are depicted as means ± standard deviation of the mean from three replicates (*n* = 3). A *p*-value of <0.05 was used to show significant differences between mean values calculated using one-way analysis of variance (ANOVA) and Duncan’s multiple-range test done using IBM^®^ SPSS^®^ Statistics version 20.0 (SPSS Inc., Chicago, IL, USA).

## 3. Results and Discussion

### 3.1. Profile of 18 Free Amino Acids in Honey Samples

The analysis of FAAs in honey samples has been based on the HPLC-ESI-TQ-MS/MS approach running under positive electrospray ionization mode to generate protonated precursor ions, followed by their collision-induced fragmentation to specific product ions. During method optimization operating under multiple reaction monitoring mode (MRM) two ion transitions for each AA were selected for quantitative and qualitative analysis. For each ion transition, the parameters such as collision energy, precursor Q1, and product Q3 voltage, and dwell time were optimized to reach sufficient chromatographic response. The MRM AA transitions, corresponding collision energy, Q1, Q3, and dwell time for investigated FAAs are given in Table 2. To ensure the reliability of the results, a detector response for particular AAs was obtained through the ordinary least squares method (OLSM), getting acceptable linearity of calibration curves in the concentration range of 0.0025–0.20 μM L^−1^. The calculated coefficients of determination (R^2^) were higher than 0.99 for all investigated AAs. Additionally, using the criterion of the signal-to-noise ratio of 3:1, limit of detection (LOD) values of 18 FAAs were achieved within 0.04 to 13.79 ng mL^−1^. While considering the criterion of a signal-to-noise ratio of 10:1, the limit of quantification (LOQ) fluctuated within the range from 0.14 to 41.78 ng mL^−1^. The relative standard deviation (RSD) of the instrumental precision (data not shown) was lower than 3.50% for relative peak area and 0.02% for retention time, indicating that the instrumental system was suitable for the analysis of FAAs without a prior derivatization step.

In the analysis of FAAs, sample extraction and purification steps play an important role as they greatly influence the recovery and correct quantification of compounds [23]. Organic and inorganic solvents such as formic acid, methanol [17], trichloroacetic acid [24], and ethyl acetate [25] are commonly used in the solid-liquid or liquid-liquid extraction of FAAs from plant matrices [25]. The yield of compounds of interest, though, could vary depending on molecular structure along with their polarity, concentration, and availability of functional groups. In consideration of environmental pollution matters and governmental intentions outlined in the EC Directive 2010/75/EU [10], aiming to reduce the negative impact of industrial toxic emissions on ecosystems, isolation along with the pre-concentration of FAAs was done supporting the protocol of Cukier et al. [21]. For this purpose, a cation-exchange column (3 mL) packed with 90.0 mg of strongly acidic gel-type fractionation resin DIAION™ UBK550 was used, while bypassing the extraction step. The results of this approach revealed no presence of either monosaccharides or disaccharides in the eluate fractions collected upon SPE (data not shown), indicating a potential utilization of it for isolation of AA from the honey matrix. The effectiveness of SPE utilizing strongly acidic cation-exchange resin Amberlite^®^ IR120 was also highlighted by Cometto et al. [26], reaching a good baseline and resolution of almost all AAs using RP-HPLC along with the derivatization method.

As seen, selective HPLC-ESI-TQ-MRM-MS/MS analysis confirmed the presence of 17 FAAs on all honey samples used in this study (Table 3). The average values of FAAs were variable depending on the type of honey and the region of origin. The highest concentration of total FAAs was observed in buckwheat flower honey from the Liepaja district (LV), the value corresponding to 98.28 ± 2.22 mg 100 g^−1^ FW. The report of Kačkeš et al. [27] highlights outstanding nutritional value of buckwheat honey as the highest FAAs content was observed compared with other honey samples. This observation could be reinforced by Rybak-Chmielewska and Szczesna [28], reporting fairly similar total FAAs content in fresh buckwheat honey. Considerably lower but still relevant concentrations of total FAAs were observed in bumblebee honey from the Altai district (RU), followed by bee polyfloral honey from Baljuvon (TJK), the values corresponded to 55.16 ± 0.91 and 40.16 ± 4.19 mg 100 g^−1^ FW, respectively. The values observed for polyfloral bee honey were consistent with those of Łozowicka et al. [17]. The lowest concentration of total FAAs was found in bee polyfloral honey from the Mazsalaca district (LV), with total amount of 22.69 ± 5.26 mg 100 g^−1^ FW.

Proline was found to be the prevalent AA (average of all samples) found in all tested honey samples. This observation is in line with Kowalski et al. [12]. The concentration of proline varied in the range of 16.50 ± 4.92–44.41 ± 0.02 mg 100 g^−1^ FW, with bumblebee honey having the highest content and polyfloral honey from the Mazsalaca district (LV) the lowest. An almost equal amount of proline was observed in buckwheat flower and bumblebee honey, however, bumblebee honey contained a statistically higher (*p* < 0.05) value. Czipa et al. [29] pointed out that the contribution of proline to the total FAAs usually ranges from 50–80%, which could serve as a criterion for detection of honey adulteration. This statement was further reinforced by Nisbet et al. [30], observing a relationship between the proline and relatively high fructose, glucose, and sucrose content. However, the proposed method seems to be less effective in detecting adulteration of buckwheat honey since, against the background of the high content of other AAs, the contribution of proline to the total amount was less pronounced (41.7%). Such as tyrosine in relatively high concentration was detected exclusively in buckwheat honey, corresponding to 15.73 ± 0.32 mg 100 g^−1^ FW. The observation in this study is consistent with findings made by Kuš (2020) [31], indicating the relative abundance of tyrosine in buckwheat honey of Polish origin; however, the observed value was 1.7-fold lower than found in this study.

A similar observation was made by Wang et al. [32], pointing out that tyrosine and formic acid could serve as markers of buckwheat pollen. A relatively high amount of tyrosine was found also in bee polyfloral honey from Baljuvon (TJK), the value corresponding to 3.31 ± 0.07 mg 100 g^−1^ FW. However, the concentration found was 3.8-fold higher and 15.3-fold lower than that observed in *Calluna vulgaris* [33] and *Vitex agnus-castus* [34] honey, respectively.

Phenylalanine was the third dominant representative of FAAs found in all five honey samples within the range from 0.84 ± 0.05 to 7.14 ± 0.42 mg 100 g^−1^ FW, with bee polyfloral honey from Baljuvon (TJK) having the highest content and bumblebee honey from Altai district (RU) the lowest. No significant difference (*p* > 0.05) was found between polyfloral (forest) and buckwheat honey samples from the Liepaja district. In the report of Kuš (2020) [31], however, the supremacy of buckwheat honey was noticed as it contained the highest amount of this AA, corresponding to 0.34 mg 100 g^−1^ FW. A fairly similar amount of this essential aromatic AA has been reported by Łozowicka et al. [17], highlighting the abundance of phenylalanine in polyfloral herbs honey from Kazakhstan over other samples originating from Poland and Belarus. Karabagias et al. [35] indicated a relatively high degree of variability in the phenylalanine and proline content in Greek honey from different geographical regions. The observations made by researchers allow hypothesizing that the formation of phenylalanine is most likely associated with a more favorable climate in regions of lower latitude than in longer.

Leucine was the fourth most abundant FAA quantified at the highest concentration of 12.52 ± 0.10 mg 100 g^−1^ FW in buckwheat flower honey (LV) and contributing 12.7% of the total amount of FAAs. Janiszewska et al. [36] reported a fairly similar concentration of leucine in buckwheat flower honey from Poland. Apart from proline, the dominance of leucine in buckwheat flower honey was also highlighted by Kortesniemi et al. [37]. The content of leucine in all other honey samples ranged from 0.36 ± 0.01 to 0.96 ± 0.08 mg 100 g^−1^ FW, with bumblebee honey from Altai district (RU) having the highest content and bee polyfloral honey from the Mazsalaca district (LV) the lowest. The observed values coincide with those of Kowalski et al. [12], showing the range of leucine from 0.75 to 1.12 mg 100 g^−1^ in polyfloral honey from Slovakia. No statistically significant difference (*p* > 0.05) in terms of leucine content was observed between honeybee polyfloral and polyfloral (forest) honey samples from Mazsalaca (LV) and Liepaja (LV) districts, respectively.

Isoleucine in almost equivalent concentrations was observed in honey samples within the range from 0.36 ± 0.04 to 12.31 ± 0.83 mg 100 g^−1^ FW. Similar to leucine, the highest concentration of this essential AA was observed in buckwheat flower honey (LV) contributing 12.5% of the total content of FAAs. The superiority of buckwheat honey over other samples was also noted by Janiszewska et al. [36], showing a value of 5.58 mg 100 g^−1^. This observation was reinforced by a nuclear magnetic resonance (NMR) analysis of the FAAs profile made by Kortesniemi et al. [37].

Apart from the major AAs observed, the concentration of aspartic acid was found to be high exceptionally in bumblebee honey, contributing up to 4.6% to the total amount of AAs. The concentration of aspartic acid in this type of honey was found to be similar to those observed by Rajindran et al. [38] for raw green and Manuka honey originating from Banggi Island, Sabah. The outstanding prevalence of aspartic acid in bumblebee honey can be used as a marker clarifying its authenticity.

Branched-chain AAs (BCAAs, valine, leucine, and isoleucine) belong to essential AAs because they are not synthesized by the human body and must be obtained from food [39]. Multiple beneficial effects of BCAAs have repeatedly been proven [40,41,42] so the importance of these compounds in the human body is undebatable. The results of this study demonstrate a fairly similar sum of BCAAs in polyfloral honey samples, the concentration varied in the range of 1.08 ± 0.08–1.28 ± 0.11 mg 100 g^−1^ FW. The lowest amount of BCAAs was statistically confirmed in polyfloral honey from Mazsalaca district, while no significant differences (*p* > 0.05) were observed between polyfloral honey samples from Baljuvon (TJK) and Liepaja district. The amount of BCAAs in buckwheat flower honey, followed by bumblebee honey stood out, corresponding to the values of 31.13 ± 1.50 and 3.17 ± 0.18 mg 100 g^−1^ FW, respectively. The results are consistent with those of Rybak-Chmielewska and Szczesna [28], observing a high concentration of BCAAs (valine and leucine) in buckwheat honey in the amount of 13.7 mg 100 g^−1^ FW. The authors also reported a relatively low concentration of isoleucine, corresponding only to 0.90 mg 100 g^−1^ FW. The total amount of BCAAs in buckwheat honey reported by Janiszewska et al. [36], corresponding to 17.6 mg 100 g^−1^, which is way closer to the values obtained in this study. It is worth noting that buckwheat honey with reasonable limits can be considered by athletes as an attractive source of BCAAs that could help reduce muscle soreness and shorten recovery time.

Due to limitations in literature data regarding the profile of FAAs in bumblebee honey, a direct comparison of observed BCAA values could not be made in this study.

### 3.2. Profile of Mono- and Disaccharides in Honey Samples

Honey adulteration, where a variety of sugar-based syrups are being used to increase bulk volume, is a distinct concern that notably affects both the honey industry and consequently leads to credibility loss by the consumers [43]. Quantitative and qualitative analysis of mono- and disaccharides to a large extent makes it possible to identify the conscientiousness of honey producers [44].

The analysis of mono- and disaccharides in honey samples revealed the presence of six sugars, with fructose and glucose making the biggest contribution to the total amount of sugars (Table 4). The observed values of fructose varied in the range of 33.05 ± 0.85–36.63 ± 0.23 g 100 g^−1^ FW, with bee polyfloral honey (TJK) having the highest content and polyfloral (forest) honey from Liepaja district (LV) the lowest. No statistically significant differences (*p* > 0.05) were observed between polyfloral (forest) honey (LV) and bumblebee honey (RU). The observed values are consistent with those of Cwiková et al. [44], indicating a fairly similar distribution of fructose among 21 honey samples. Glucose was the second prevalent monosaccharide observed in all honey samples within the range of 24.05 ± 0.79–28.11 ± 0.10 g 100 g^−1^ FW. Similar to fructose, the highest content of glucose was observed in polyfloral honey (TJK), while the lowest in polyfloral (forest) and polyfloral honey originated from Liepaja and Mazsalaca districts (LV), respectively.

Considering the sum of fructose and glucose, which should be not less than 60 g 100 g^−1^ FW, only two out of five honey samples complained with the criteria outlined in the Council Directive 2001/110/EC [10]. The sum of fructose and glucose was the highest in bee polyfloral honey from Baljuvon (TJK) and buckwheat flower honey (LV), corresponding to the values of 64.74 ± 0.33 and 63.59 ± 1.79 g 100 g^−1^ FW, respectively. The lowest sum of fructose and glucose was found in bumblebee honey (RU), followed by polyfloral (forest) honey samples from Liepaja (LV), the value corresponded to 56.88 ± 1.89 and 57.97 ± 1.49 g 100 g^−1^ FW, respectively.

The statement made by Islam et al. [43], discloses that usually high content of either sucrose or maltose indicates possible honey adulteration. It is worth noting, though, that neither sucrose nor maltose was detected in any of the honey samples, which is in line with the observation made by Jiang et al. [45]. All honey samples, except for buckwheat showed the presence of four unknown compounds. The report of Amariei et al. [46] demonstrated that trehalose apart from glucose and fructose was the third most prevalent sugar found in the honey of various origins and could contribute up to 2.26% to the total content of sugars. As part of the supplementary study, comparing the retention times of the unknown peaks detected in honey samples with those of trehalose standard revealed no presence of this sugar in none of the samples. Meanwhile, the mentioned unknown minor sugars (unknown peaks 1, 2, 3, and 4) have not been confidently identified, so they are not discussed in this study. Presumably, unidentified sugars may be representatives of oligosaccharides or sugar alcohols as proposed by Jiang et al. [45], though an additional study is needed.

## 4. Conclusions

The developed and utilized methods for preparation and quantitative analysis of 18 FAAs in honey in a shorter time and easier way. Simple clean-up and pre-concentration step using cation-exchange DIAION™ UBK550 resin delivered comparable with literature data FAAs values while ensuring the absence of sugars negatively affecting ionization efficiency along with reducing the signal intensity. The HPLC-ESI-TQ-MS/MS system was demonstrated to be reliable for the detection and quantification of FAAs in five honey samples bypassing the derivatization process and ensuring high sensitivity with limits of detection and quantification ranging from 0.04 to 13.79 ng mL^−1^ and from 0.14 to 41.78 ng mL^−1^, respectively. Quantitative analysis revealed the presence of 17 FAAs in all honey samples, with proline being the prevalent representative contributing to the total amount of FAAs within the range from 41.7% to 80.52%. The highest concentration of proline was found in bumblebee honey, followed by buckwheat flower honey originating from Russia and Latvia, respectively. While the lowest amount of proline was observed in polyfloral honey from Latvia, corresponding to 16.50 mg 100 g^−1^ FW. Relatively low content of proline in polyfloral honey from the Mazsalaca district (LV) led to speculating on possible adulterations of this sample by substituting part of harvest honey with exogenous sugars, in particular sucrose and maltose. However, further analysis of individual sugars revealed the presence of neither sucrose nor maltose in any of the honey samples. Considering the sum of glucose and fructose, only two out of five honey samples complained with the criteria outlined in the Council Directive 2001/110/EC. The results of this study demonstrated substantial superiority of buckwheat honey in terms of tyrosine, leucine, and isoleucine content, leading to propose these three FAAs to be used as markers in the identification of honey type solely based on AAs content. Meanwhile, the abundance of branched-chain AAs both in buckwheat and bumblebee honey makes these types of honey unique from a nutritional standpoint of view, as they could ensure organism with essential AAs and promote muscle growth and alleviate muscle soreness relevant to athletes.

## Figures and Tables

**Figure 1 foods-11-02744-f001:**
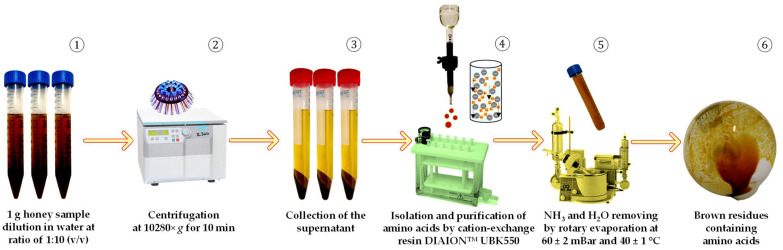
A schematic representation of the sample preparation steps undertaken for the extraction and purification of amino acids present in honey matrix.

**Figure 2 foods-11-02744-f002:**
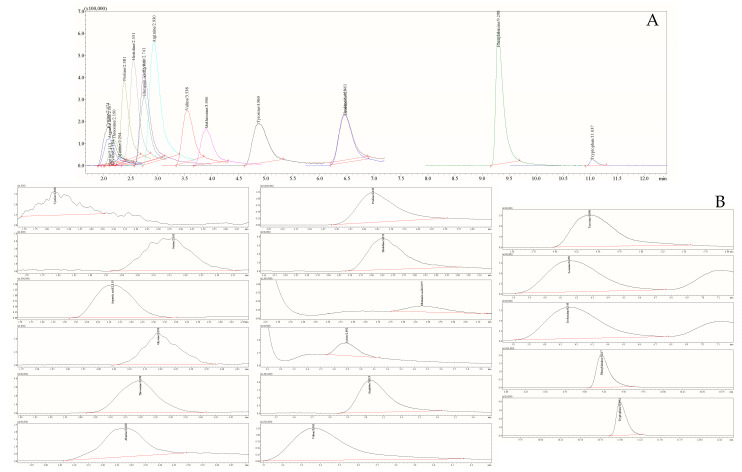
Total ion current (TIC) chromatograms in MRM mode are given for 18 amino acid standards at the concentration of 0.078 μM L^−1^ (**A**) and honey sample (**B**). Samples injection volume 3.0 μL (0.003 μg mL^−1^).

**Figure 3 foods-11-02744-f003:**
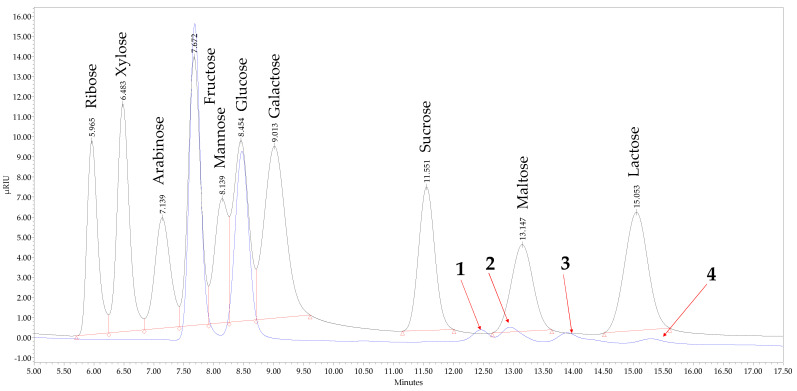
Representative chromatographic separation of mono- and disaccharide standards (black line) and sugars detected in honey samples (blue line). Samples injection volume 15.0 μL (0.075 μg mL^−1^). Peaks 1, 2, 3, and 4 refer to not identified compounds detected in honey samples with the retention time 12.447, 12.935, 13.884, and 15.286 min, respectively.

**Table 1 foods-11-02744-t001:** Detailed characteristics of honey samples subjected to analysis.

Honey Type	Origin	Year of Collection	Hive Location
Polyfloral, bee	TJK	2020	Tajikistan, Baljuvon (38°18′30.0″ N 69°40′35.0″ E)
Polyfloral, bee	LV	2020	Latvia, Mazsalaca district, (57°51′47.7″ N 25°01′32.8″ E)
Polyfloral (forest), bee	LV	2019	Latvia, Liepaja district, Gramzda parish (56°22′00.8″ N 21°36′24.7″ E)
Buckwheat, bee	LV	2019	Latvia, Liepaja district, Gramzda parish (56°22′00.8″ N 21°36′24.7″ E)
Polyfloral, bumblebee	RU	2021	Russia, Artybash village (51°47′25.8″ N 87°15′13.7″ E)

**Table 2 foods-11-02744-t002:** The MRM transitions and corresponding collision energy, Q1, Q3 and dwell time for investigated amino acid compounds.

Optimized MRM Parameters									Parameters of Calibration		
Amino Acid	Rt, min	Molecular Formula	Ionization Mode	MRM Transitions	Q1 Pre Bias, V	Collision Energy, V	Q3 Pre Bias, V	Dwell Time, Msec	R^2^	LOD, ng mL^−1^	LOQ, ng mL^−1^
Cysteine	2.074	C_6_H_12_N_2_O_4_S_2_	[M + H]^+^	241.1000→74.0000	15.0	27.0	16.0	15.0	0.9988	5.25	15.95
				241.1000→152.0500	15.0	15.0	18.0	15.0			
Aspartic acid	2.107	C_4_H_7_NO_4_	[M + H]^+^	134.1000→73.9500	14.0	15.0	16.0	15.0	0.9992	3.50	10.60
				134.1000→88.1000	20.0	11.0	20.0	15.0			
Serine	2.113	C_3_H_7_NO_3_	[M + H]^+^	105.4000→60.0000	17.0	12.0	20.0	34.0	0.9989	13.79	41.78
				105.4000→50.0000	18.0	19.0	21.0	34.0			
Glycine	2.164	C_2_H_5_NO_2_	[M + H]^+^	76.3000→30.0500	11.0	14.0	25.0	34.0	0.9990	6.06	18.37
Threonine	2.190	C_4_H_9_NO_3_	[M + H]^+^	120.1000→74.0500 120.1000→56.1500	13.0 13.0	11.0 15.0	17.0 23.0	15.0 15.0	0.9995	4.35	13.17
Alanine	2.294	C_3_H_7_NO_2_	[M + H]^+^	90.0000→44.1000 90.0000→50.1000	10.0 9.0	12.0 15.0	19.0 21.0	34.0 34.0	0.9993	3.43	10.39
Proline	2.381	C_5_H_9_NO_2_	[M + H]^+^	116.0000→70.1500 116.0000→28.0500	17.0 16.0	16.0 39.0	15.0 12.0	15.0 15.0	0.9997	2.33	7.05
Histidine	2.551	C_6_H_9_N_3_O_2_	[M + H]^+^	156.1000→110.1000 156.1000→56.1000	22.0 23.0	15.0 32.0	13.0 13.0	15.0 15.0	0.9991	7.22	21.88
Lysine	2.741	C_6_H_14_N_2_O_2_	[M + H]^+^	147.1000→84.1000 147.1000→130.1000	16.0 15.0	18.0 24.0	19.0 15.0	34.0 34.0	0.9992	6.00	18.19
Glutamic acid	2.747	C_5_H_9_NO_4_	[M + H]^+^	147.4000→84.1000 147.4000→56.1000	16.0 17.0	17.0 30.0	19.0 24.0	15.0 15.0	0.9990	7.33	22.21
Arginine	2.930	C_6_H_14_N_4_O_2_	[M + H]^+^	175.1000→70.1000 175.1000→60.1000	20.0 19.0	23.0 14.0	15.0 13.0	15.0 15.0	0.9992	6.72	20.37
Valine	3.538	C_5_H_11_NO_2_	[M + H]^+^	118.1000→72.1500 118.1000→55.0500	18.0 17.0	12.0 22.0	15.0 12.0	34.0 34.0	0.9989	5.01	15.19
Methionine	3.900	C_5_H_11_NO_2_S	[M + H]^+^	150.1000→56.0000 150.1000→104.1000	16.0 16.0	16.0 12.0	12.0 13.0	34.0 34.0	0.9989	5.19	15.73
Tyrosine	4.860	C_9_H_11_NO_3_	[M + H]^+^	182.1000→135.9000 182.1000→91.1000	27.0 28.0	13.0 28.0	16.0 21.0	34.0 34.0	0.9997	3.12	9.45
Isoleucine	6.461	C_6_H_13_NO_2_	[M + H]^+^	132.1000→86.1000 132.1000→69.0500	20.0 19.0	11.0 18.0	19.0 16.0	59.0 59.0	0.9992	4.79	14.52
Leucine	6.465	C_6_H_13_NO_2_	[M + H]^+^	132.3000→86.1500 132.1000→30.0500	16.0 20.0	12.0 17.0	19.0 15.0	59.0 59.0	0.9997	2.22	6.72
Phenylalanine	9.298	C_9_H_11_NO_2_	[M + H]^+^	166.1000→120.1000 166.1000→103.1000	25.0 18.0	14.0 26.0	14.0 13.0	59.0 59.0	0.9996	1.48	4.48
Tryptophan	11.037	C_11_H_12_N_2_O_2_	[M + H]^+^	205.0000→188.1500 205.0000→146.1000	14.0 22.0	11.0 20.0	15.0 17.0	90.0 90.0	0.9998	0.04	0.14

Note: The first MRM amino acid transitions found were used for quantitative analysis, while the second for qualitative. LOD—limit of detection; LOQ—limit of quantification; RT—retention time; ng—nanogram.

**Table 3 foods-11-02744-t003:** The concentration of free amino acids in selected honey samples.

Amino Acid	Mw	Average Concentration, mg 100 g^−1^ FW				
		1	2	3	4	5
Aspartic acid	133.10	0.07 ± 0.00 ^d^	0.17 ± 0.02 ^c^	0.17 ± 0.00 ^c^	0.31 ± 0.00 ^b^	2.46 ± 0.17 ^a^
Cysteine	240.30	<LOQ	<LOQ	<LOQ	0.04 ± 0.00	<LOQ
Serine	105.09	0.16 ± 0.01 ^e^	0.18 ± 0.08 ^d^	0.26 ± 0.02 ^c^	1.29 ± 0.04 ^a^	0.89 ± 0.03 ^b^
Threonine	119.12	0.10 ± 0.00 ^d^	0.12 ± 0.00 ^d^	0.16 ± 0.00 ^c^	1.08 ± 0.02 ^a^	0.42 ± 0.02 ^b^
Glycine	75.07	0.13 ± 0.01 ^c^	0.13 ± 0.00 ^c^	0.14 ± 0.01 ^c^	0.63 ± 0.02 ^a^	0.30 ± 0.01 ^b^
Proline	115.13	26.50 ± 2.97 ^c^	16.50 ± 4.92 ^e^	24.38 ± 0.42 ^d^	41.02 ± 0.23 ^b^	44.41 ± 0.02 ^a^
Alanine	89.09	0.24 ± 0.26 ^e^	0.55 ± 0.01 ^d^	0.59 ± 0.01 ^c^	1.07 ± 0.01 ^a^	0.74 ± 0.02 ^b^
Histidine	155.15	0.14 ± 0.04 ^a,b^	0.03 ± 0.01 ^c^	0.11 ± 0.01 ^b^	0.15 ± 0.01 ^a^	0.15 ± 0.00 ^a^
Lysine	146.19	0.52 ± 0.17 ^b^	0.29 ± 0.02 ^c^	0.51 ± 0.02 ^b^	0.68 ± 0.01 ^a^	0.12 ± 0.05 ^d^
Glutamic acid	147.13	0.51 ± 0.14 ^b^	0.27 ± 0.02 ^d^	0.47 ± 0.03 ^c^	0.65 ± 0.02 ^a^	0.13 ± 0.06 ^e^
Arginine	174.20	0.10 ± 0.03 ^d^	0.10 ± 0.00 ^d^	0.27 ± 0.01 ^b^	0.19 ± 0.01 ^c^	0.78 ± 0.07 ^a^
Valine	117.15	0.39 ± 0.00 ^d^	0.35 ± 0.03 ^e^	0.45 ± 0.06 ^c^	6.30 ± 0.57 ^a^	1.21 ± 0.02 ^b^
Methionine	149.21	n.d.	n.d.	n.d.	n.d.	n.d.
Tyrosine	181.19	3.31 ± 0.07 ^b^	0.41 ± 0.04 ^e^	0.60 ± 0.01 ^d^	15.73 ± 0.32 ^a^	0.73 ± 0.23 ^c^
Leucine	131.17	0.43 ± 0.01 ^c^	0.36 ± 0.01 ^d^	0.38 ± 0.04 ^d^	12.52 ± 0.10 ^a^	0.96 ± 0.08 ^b^
Isoleucine	131.17	0.45 ± 0.02 ^c^	0.36 ± 0.04 ^d^	0.45 ± 0.01 ^c^	12.31 ± 0.83 ^a^	1.00 ± 0.08 ^b^
Phenylalanine	165.19	7.14 ± 0.42 ^a^	2.87 ± 0.06 ^d^	4.78 ± 0.05 ^b^	4.31 ± 0.03 ^b^	0.84 ± 0.05 ^c^
Tryptophan	204.23	0.00 ± 0.00	0.00 ±0.00	0.01 ± 0.00 ^a^	0.00 ± 0.00	0.02 ± 0.00 ^a^
**∑_BCAA_**	**−**	**1.27 ± 0.03 ^c^**	**1.08 ± 0.08 ^d^**	**1.28 ± 0.11 ^c^**	**31.13 ± 1.50 ^a^**	**3.17 ± 0.18 ^b^**
**∑_total_**	**−**	**40.16 ± 4.19 ^c^**	**22.69 ± 5.26 ^e^**	**33.73 ± 0.70 ^d^**	**98.28 ± 2.22 ^a^**	**55.16 ± 0.91 ^b^**

Note: Values are means ± SD of triplicates (*n* = 3). LOQ—limit of quantification; BCAA—branched-chain amino acid; n.d.—not detected; FW—fresh weight basis. Honey samples of various origins and regions selected for analysis of amino acids were: 1—polyfloral (Tajikistan); 2—polyfloral (Latvia); 3—polyfloral (forest) (Latvia); 4—buckwheat flower (Latvia); 5—bumblebee (Russia). Means within the same amino acid with different superscript letters (^a,b,c,d,e^) are significantly different at *p* < 0.05.

**Table 4 foods-11-02744-t004:** The concentration of individual sugars in selected honey samples, g 100^−1^.

Sugar	RT, min	Average Concentration, g 100 g^−1^ FW				
		1	2	3	4	5
Fructose	7.62	36.63 ± 0.23 ^a^	35.20 ± 2.90 ^b^	33.05 ± 0.85 ^c^	36.05 ± 1.16 ^a^	32.83 ± 1.10 ^c^
Glucose	8.454	28.11 ± 0.10 ^a^	24.74 ± 3.08 ^b^	24.92 ± 0.64 ^b^	27.54 ± 0.63 ^a^	24.05 ± 0.79 ^b^
Sucrose	11.551	n.d.	n.d.	n.d.	n.d.	n.d.
Unknown 1	12.447	1.49 ± 0.03 ^a^	1.57 ± 0.12 ^a^	1.56 ± 0.07 ^a^	1.26 ± 0.02 ^b^	1.61 ± 0.04 ^a^
Unknown 2	13.935	1.47 ± 0.16 ^b^	2.34 ± 0.31 ^a^	2.32 ± 0.02 ^a^	0.88 ± 0.04 ^c^	1.46 ± 0.03 ^b^
Unknown 3	13.884	0.84 ± 0.03 ^c^	1.60 ± 0.13 ^a^	1.68 ± 0.01 ^a^	0.86± 0.07 ^c^	1.46 ± 0.03 ^b^
Unknown 4	15.286	0.86 ± 0.00 ^c^	1.25 ± 0.19 ^b^	1.45 ± 0.07 ^a^	n.d.	1.24 ± 0.02 ^b^
**∑_total_**	**−**	**69.40 ± 0.55 ^a^**	**66.70 ± 6.73 ^b^**	**64.98 ± 1.66 ^c^**	**66.59 ± 1.92 ^b^**	**62.65 ± 2.01 ^d^**

Note: Values are means ± SD of triplicates (*n* = 3). Quantitative analysis of the unknown compounds (third, fourth, fifth, and sixth peak of the chromatogram—Figure 2) was performed according to the fructose calibration curve. FW—fresh weight basis; RT—retention time; n.d.—not detected. Honey samples of various origins and regions selected for analysis of amino acids were: 1—polyfloral (Tajikistan); 2—polyfloral (Latvia); 3—polyfloral (forest) (Latvia); 4—buckwheat flower (Latvia); 5—bumblebee (Russia). Means within the same sugar with different superscript letters (^a,b,c,d^) are significantly different at *p* < 0.05.

## Data Availability

Data are contained within the article.

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
