# Peer review of "Analysis of 18 Free Amino Acids in Honeybee and Bumblebee Honey from Eastern and Northern Europe and Central Asia Using HPLC-ESI-TQ-MS/MS Approach Bypassing Derivatization Step"

_foods, 2022, doi:10.3390/foods11182744_

Round 1

Reviewer 1 Report

Analysis of 18 Free Amino Acids in Bee and Bumblebee Honey from Eastern and Northern Europe and Central Asia using HPLC-ESI-TQ-MS/MS Approach Bypassing Derivatization Step

The aim of the presented work is very interesting and the authors provided an experimental data showing the Analysis of 18 Free Amino Acids in different honeys (bumblebee honey is especially interesting) from Eastern and Northern Europe and Central Asia.

Some suggestions for Authors:

Line 1: Since it is conventional honey, I assume that it is from Apis mellifera L., and I suggest that the word honey be inserted in the title instead of the word bee, so the title would be...Analysis of 18 Free Amino Acids in Honeybee and Bumblebee Honey from Eastern and Northern Europe and Central Asia using HPLC-ESI-TQ-MS/MS Approach Bypassing Derivatization Step

Line 4: after the last name there must be a space and then the affiliation number

Line 14: please correct

Line 15: the abstract is too long – according the instructions for authors the maximum is 200 words. Please correct

Line 94: Honey samples not Plant material because even though honey comes from a plants, it is an animal product. Therefore, it cannot be said that you worked on plant material

Line 100: Table 1. The honey samples are not adequately declared. According to the International Honey Commission (IHC), there is multifloral or polyfloral honey. What is Wildflower honey? It is multifloral honey, so please change it. Also, buckwheat honey. Has it been confirmed by melissopalynological analysis and physical-chemical analyses? Also, in this table sample Polyfloral, bee sample LV 2020 GPS coordinate is missing.

Lines 121, 132, 150: Correct into H2O

Line 270: Italic style for Calluna vulgaris and Vitex agnus-castus

Line 271: delete word flowers

Line 355: make space before line 356

Line 414: delete Appendix A

Line 415. Please make a list of references according to the instructions for authors:

Example: Al-Waili, N.S.; Haq, A. Effect of Honey on Antibody Production against Thymus-Dependent and Thymus-Independent Antigens in Primary and Secondary Immune Responses. J. Med. Food 2004, 7, 491–494. doi:10.1089/jmf.2004.7.491.

Author Response

Response to Reviewer’s 1 comments

R: Analysis of 18 Free Amino Acids in Bee and Bumblebee Honey from Eastern and Northern Europe and Central Asia using HPLC-ESI-TQ-MS/MS Approach Bypassing Derivatization Step

R: The aim of the presented work is very interesting and the authors provided an experimental data showing the Analysis of 18 Free Amino Acids in different honeys (bumblebee honey is especially interesting) from Eastern and Northern Europe and Central Asia.

R: Some suggestions for Authors:

A: The authors would like to thank the reviewer for carefully checking our manuscript and for valuable comments. In preparing the manuscript authors have incorporated most of the changes suggested. The authors refer to them in detail below.

R: Line 1: Since it is conventional honey, I assume that it is from Apis mellifera L., and I suggest that the word honey be inserted in the title instead of the word bee, so the title would be...Analysis of 18 Free Amino Acids in Honeybee and Bumblebee Honey from Eastern and Northern Europe and Central Asia using HPLC-ESI-TQ-MS/MS Approach Bypassing Derivatization Step

A: The authors appreciate the reviewer's suggestion very much. The title of the work has been changed.

R: Line 4: after the last name there must be a space and then the affiliation number

A: This shortcoming has been corrected.

R: Line 14: please correct             

A: The authors removed the repeated word “Correspondence”

R: Line 15: the abstract is too long – according the instructions for authors the maximum is 200 words. Please correct

A: The authors thank the reviewer for his valuable remark. The amount of words in the Abstract section has been reduced to 206 words.

R: Line 94: Honey samples not Plant material because even though honey comes from a plants, it is an animal product. Therefore, it cannot be said that you worked on plant material

A: The authors thank the reviewer for his valuable remark. The phrase “Plant Material” was substituted with “Honey Samples”.

R: Line 100: Table 1. The honey samples are not adequately declared. According to the International Honey Commission (IHC), there is multifloral or polyfloral honey. What is Wildflower honey? It is multifloral honey, so please change it. Also, buckwheat honey. Has it been confirmed by melissopalynological analysis and physical-chemical analyses? Also, in this table sample Polyfloral, bee sample LV 2020 GPS coordinate is missing.

A: The authors agree with the reviewer’s point and they are sorry for misleading the reviewer. The word “wildflower” was used since according to the information provided by the manufacturer, this honey was harvested from wild herbs in the Latvian forest. The authors changed the word “wildflower” to “polyfloral” with a note "forest".

Unfortunately, neither melissopalynological nor physical-chemical analyses were done to identify the origin of the honey investigated. The authors rely solely on the manufacturer's information, taking his reputation and well-recognition in the Latvia market as a basis.

R: Lines 121, 132, 150: Correct into H2O

A: Corrected.

R: Line 270: Italic style for Calluna vulgaris and Vitex agnus-castus

A: Corrected.

R: Line 271: delete word flowers

A: The word “flowers” has been deleted.

R: Line 355: make space before line 356

A: Corrected.

R: Line 414: delete Appendix A

A: The “Appendix A” has been deleted.

R: Line 415. Please make a list of references according to the instructions for authors:

Example: Al-Waili, N.S.; Haq, A. Effect of Honey on Antibody Production against Thymus-Dependent and Thymus-Independent Antigens in Primary and Secondary Immune Responses. J. Med. Food 2004, 7, 491–494. doi:10.1089/jmf.2004.7.491.

A: The authors thank the reviewer for his valuable remark. A list of references has been revised accordingly.

On behalf of all the co-authors

Yours sincerely,

Vitalijs Radenkovs

Principal investigator, Latvia University of Life Sciences and Technologies, Research Laboratory of Biotechnology, Division of Smart Technologies.

Reviewer 2 Report

The manuscript investigated the analysis of Free Amino Acids in Bee and Bumblebee Honey from Eastern and Northern Europe and Central Asia using HPLC-ESI-TQ-MS/MS Approach. However, the scientific significance of this paper is poor, and I did not find the novelty of this study. 

Author Response

Response to Reviewer’s 2 comments

R: The manuscript investigated the analysis of Free Amino Acids in Bee and Bumblebee Honey from Eastern and Northern Europe and Central Asia using HPLC-ESI-TQ-MS/MS Approach. However, the scientific significance of this paper is poor, and I did not find the novelty of this study.

A: The authors are sorry the reviewer spent his time reading this work. However, as an argument against the statement about the lack of novelty, the authors wish to oppose the reviewer. This manuscript discusses the composition of amino acids and individual sugars in bumblebee honey which from ancient times believed to possess health-promoting benefits. Collecting the nectar from bumblebee nests is an activity that has been practiced within living memory in many parts of the Nordic countries.  “Bumblebee honey” or “bumblebee mead” produced by bumblebees has been used as a sweet substance long before apiculture was introduced with the arrival of Christianity in the Nordic countries. There are data evidencing that bumblebee honey was used as a folk remedy in treating such ailments as inflammation (data on otitis media) or for wound treatment in combination with liqueur. Nonetheless, there is absolutely no information to date, regarding the nutritional composition of this according to folk medicine unique product. The authors provided only part of their research focusing on the content of amino acids and mono- and disaccharides in bumblebee honey as these major constituents are well studied in conventional bee honey. However, the authors will try to give more detailed information as part of subsequent works.

We believe that the results will be of interest to the readership of the journal Foods.

On behalf of all the co-authors

Yours sincerely,

Vitalijs Radenkovs

Principal investigator, Latvia University of Life Sciences and Technologies, Research Laboratory of Biotechnology, Division of Smart Technologies.

Reviewer 3 Report

Review of the manuscript entitled: "Analysis of 18 Free Amino Acids in Bee and Bumblebee Honey from Eastern and Northern Europe and Central Asia using HPLC-ESI-TQ-MS/MS Approach Bypassing Derivatization Step".

The manuscript presents the free amino acid content and sugars in several kinds of honey produced by bees and bumblebees from Europe and Asia. Authors investigated the polyfloral bee honey from Tajikistan (sample 1) and Latvia (sample 2), wildflower bee honey from Latvia (sample 3), buckwheat bee honey from Latvia (sample 4), and polyfloral, bumblebee honey from Russia (sample 5). The solid phase extraction with cation-exchange resin DIAION™ UBK550 was used to isolate and purify free amino acids from the remaining components of honey (mainly sugars). At the same time, the liquid chromatography coupled with triple MS systems was used for free amino acid quantification. The authors' work is vast, but some information is unclear to the reader. They are mostly connected with the methodology of the experiments. 

The abstract contains the suggestion (line 19) that to isolate FAA ion chromatography was applied while this process was performed by solid phase extraction (SPE).

The applied apparatus is Shimadzu Nexera UC (see part equipment 2.4). It combines supercritical fluid extraction with supercritical fluid chromatography (SFE-SFC). It is also joined with MS/MS system. Thus, according to the manufacturer, this equipment should allow the isolation and purification part to be limited from the determination process to one step. However, the SFC conditions for the Nexera apparatus are not sufficiently presented. Additionally, the Authors do not satisfactorily stress why they applied an additional SPE system to determine FAA. Readers may find the suggestion that sugars and matrix interfere with FAA, and such a process does not allow for the simultaneous determination of both class compounds and standalone FAA.

The manuscript title should indicate that the SFE-SFC system coupled with ESI-TQ-MS/MS detection was used (or Nexera UC ESI-TQ-MS/MS) for the FAA determination. Similar equipment was used by Radenkovs and his team [see 22]. Thus, why is plenty of information about HPLC-MS in the manuscript?

The procedure of the FAA extraction was based on Cuker and co-workers [ref. 21], but it differs in the extraction mixture. In 21, it was a mixture of methanol/chloroform/water, while in the manuscript, it was only water. Also, the type of SPE cation-exchanger resin applied varies. Thus, the sentence presenting that extraction was following the protocol [21] with minor modifications should be changed (line 118) ...based on [21]. The unit of pressure in line 129 of the manuscript should be mm Hg, not "Hg.

The free amino acid determination part in the manuscript is a little confusing. All samples (1-5) contained the highest level of proline. But the content and order of the remaining FAA amino acids differ. Thus, phenylalanine is the second for samples (1-3), but for second for sample 4 is tyrosine, while the second in 5 is aspartic acid. Therefore, the Authors should indicate that the content of the remaining FAA may identify honey kind (e.g., as the fingerprint). Authors should also add in the text that bumblebee honey contains a high level of aspartic acid compared to the remaining samples. The reviewer found such information only in the Abstract.

The separation of saccharides was determined by HPLC- RID technique, but there is no information about the kind of column applied to this purpose. The profiles of mono and disaccharides are presented in Fig. 3 (as chromatograms). It superimposes chromatograms of two mixtures, one stock and the honey sample. The latter one possesses two not identified peaks, 2 and 4. However, looking at stock and honey mixtures chromatograms, these not identified zones may be maltose and lactose since the localization of the chromatogram's peaks may depend on the composition and concentration of the sample components.
On the other hand, why was the MS/MS system not used for sugar peak identification? The authors used such detection for FAA. Which honey sample was presented as the blue line chromatogram in Fig. 3? The Authors investigated five samples.

The information about the sugars in honey is presented only in the main text, not in the abstract (the reviewer knows there is a word limitation regarding the Abstract). However, it also should be presented in the keywords. Also, the abbreviation SFE-SFC-ESI-TQ-MS/MS should be introduced to this part. 

The above-presented comments did not diminish the great work done during experiments but clarified the manuscript for the reader. 

Author Response

Response to Reviewer’s 3 comments

R: Review of the manuscript entitled: "Analysis of 18 Free Amino Acids in Bee and Bumblebee Honey from Eastern and Northern Europe and Central Asia using HPLC-ESI-TQ-MS/MS Approach Bypassing Derivatization Step".

R: The manuscript presents the free amino acid content and sugars in several kinds of honey produced by bees and bumblebees from Europe and Asia. Authors investigated the polyfloral bee honey from Tajikistan (sample 1) and Latvia (sample 2), wildflower bee honey from Latvia (sample 3), buckwheat bee honey from Latvia (sample 4), and polyfloral, bumblebee honey from Russia (sample 5). The solid phase extraction with cation-exchange resin DIAION™ UBK550 was used to isolate and purify free amino acids from the remaining components of honey (mainly sugars). At the same time, the liquid chromatography coupled with triple MS systems was used for free amino acid quantification. The authors' work is vast, but some information is unclear to the reader. They are mostly connected with the methodology of the experiments.

A: The authors would like to thank the Reviewer for carefully checking our manuscript and for valuable comments. In preparing the manuscript authors have incorporated most of the changes suggested. The authors refer to them in detail below.

R: The abstract contains the suggestion (line 19) that to isolate FAA ion chromatography was applied while this process was performed by solid phase extraction (SPE).

A: The authors thank the Reviewer for his valuable remark. The Abstract section has been revised along with omitting the phrase “ion chromatography”.

R: The applied apparatus is Shimadzu Nexera UC (see part equipment 2.4). It combines supercritical fluid extraction with supercritical fluid chromatography (SFE-SFC). It is also joined with MS/MS system. Thus, according to the manufacturer, this equipment should allow the isolation and purification part to be limited from the determination process to one step. However, the SFC conditions for the Nexera apparatus are not sufficiently presented. Additionally, the Authors do not satisfactorily stress why they applied an additional SPE system to determine FAA. Readers may find the suggestion that sugars and matrix interfere with FAA, and such a process does not allow for the simultaneous determination of both class compounds and standalone FAA.

A: The authors agree with the Reviewer’s point regarding the capabilities of SFE-SFC. This system does allow the extraction of compounds of interest using supercritical conditions of fluids such as CO2. The authors are familiar with several works published so far that demonstrated the successful utilization of such systems in bioactives extraction from various matrices, including honey. However, the authors would like to point out that in this experiment a CO2 for the extraction of amino acids from honey due to contamination of the system was not used. In the attempt to extract the amino acids from the honey matrix due to the chemical nature of amino acids (present as polar and non-polar), the authors were forced to use co-solvents such as water or methanol to enhance the solubilization properties of CO2. Along with adding more polar solvents to the CO2, the authors encountered severe contamination of the whole system, manifested in clogging HPLC capillaries, delivery pumps, and desolvation line. Given the complexity of honey and the availability of sugars that affect ionization efficiency and reduce the signal intensity, it was decided to include a purification step to minimize contamination of the LC-MS system and omit the matrix effect (signal/ion suppression). 

A: Dear Reviewer. To not mislead the readers, the authors omitted these two words, i.e., SFC-SFE in the manuscript.

R: The manuscript title should indicate that the SFE-SFC system coupled with ESI-TQ-MS/MS detection was used (or Nexera UC ESI-TQ-MS/MS) for the FAA determination. Similar equipment was used by Radenkovs and his team [see 22]. Thus, why is plenty of information about HPLC-MS in the manuscript?

A: In an attempt to extract amino acids from honey samples using SPS, the authors did not achieve sufficient results to be reported. This approach was not used in this study. The emphasis in this work was on optimizing the detection conditions using as HPLC-ESI-TQ-MS/MS approach, therefore so plenty of information is in this manuscript on this approach.

R: The procedure of the FAA extraction was based on Cuker and co-workers [ref. 21], but it differs in the extraction mixture. In 21, it was a mixture of methanol/chloroform/water, while in the manuscript, it was only water. Also, the type of SPE cation-exchanger resin applied varies. Thus, the sentence presenting that extraction was following the protocol [21] with minor modifications should be changed (line 118) ...based on [21].

A: The authors are grateful for the valuable remark. This fragment was amended.

R: The unit of pressure in line 129 of the manuscript should be mm Hg, not "Hg.

A: The authors are grateful for the valuable remark. A double quotation mark refers to inches of mercury, the pressure unit displayed on the manometer of the SPE system.

R: The free amino acid determination part in the manuscript is a little confusing. All samples (1-5) contained the highest level of proline. But the content and order of the remaining FAA amino acids differ. Thus, phenylalanine is the second for samples (1-3), but for second for sample 4 is tyrosine, while the second in 5 is aspartic acid. Therefore, the Authors should indicate that the content of the remaining FAA may identify honey kind (e.g., as the fingerprint). Authors should also add in the text that bumblebee honey contains a high level of aspartic acid compared to the remaining samples. The reviewer found such information only in the Abstract.

A: The order of amino acids prevalence was based on average values that were calculated for a respective amino acid in all five samples. Meanwhile, in each case, the authors provided detailed information on which sample contains more and which less a particular amino acid.

The authors appreciate a valuable suggestion regarding the inclusion of additional information stressing the high content of aspartic acid in bumblebee honey. An additional paragraph (Lines 311-316) has been introduced highlighting the prevalence of aspartic acid in bumblebee honey.

R: The separation of saccharides was determined by HPLC- RID technique, but there is no information about the kind of column applied to this purpose.

A: Additional information on analytical conditions was ensured in Section “2.6. The HPLC-RID Conditions for Carbohydrates Analysis

R: The profiles of mono and disaccharides are presented in Fig. 3 (as chromatograms). It superimposes chromatograms of two mixtures, one stock and the honey sample. The latter one possesses two not identified peaks, 2 and 4. However, looking at stock and honey mixtures chromatograms, these not identified zones may be maltose and lactose since the localization of the chromatogram's peaks may depend on the composition and concentration of the sample components.

A: The authors had a similar opinion about not identifying peaks. However, in a separate analysis with the injection of a higher sample amount, the authors observed a similar shift of unknown peaks. The difference in retention time was noted exclusively for unknown peaks rather than for glucose and fructose, which makes identification doubtful.

R: On the other hand, why was the MS/MS system not used for sugar peak identification? The authors used such detection for FAA.

A: Authors in their research applied multiple reaction monitoring (MRM) using triple quadrupole mass selective detection HPLC-ESI-TQ-MS/MS. This approach is highly selective and sensitive allowing to work with targeted metabolomics within profiling and quantitative analysis. However, it is less effective in non-targeted metabolomics since identification is based on scheduled MRM transitions (Q1→Q3) obtained during system optimization dealing with external analytical standards. To the best of our knowledge, LC systems coupled with high-resolution mass spectrometry (LC-HRMS) or quadrupole time-of-flight mass spectrometry (Q-TOF) for non-targeted metabolite profiling due to high resolution deliver very reliable and accurate results. 

R: Which honey sample was presented as the blue line chromatogram in Fig. 3? The Authors investigated five samples.

A: The blue line presented in Figure 3 corresponds to the profile of sugars found in sample 2 (Mazsalaca district). However, since the profile of sugars was found to be similar in all five honey samples, the authors decided to not include all five chromatograms.

R: The information about the sugars in honey is presented only in the main text, not in the abstract (the reviewer knows there is a word limitation regarding the Abstract). However, it also should be presented in the keywords. Also, the abbreviation SFE-SFC-ESI-TQ-MS/MS should be introduced to this part.

The above-presented comments did not diminish the great work done during experiments but clarified the manuscript for the reader.

A: The authors are grateful for the substantiated remark. In the abstract and keywords, the authors mentioned about the analysis of mono and disaccharides carried out in this research.

On behalf of all the co-authors

Yours sincerely,

Vitalijs Radenkovs

Principal investigator, Latvia University of Life Sciences and Technologies, Research Laboratory of Biotechnology, Division of Smart Technologies.

Round 2

Reviewer 3 Report

Dear authors

The authors figured in the reviewer's suggestion. Additionally, the answers to the reviewer's questions are satisfactory. Therefore, the manuscript value is improved and may be published in the present form.

Author Response

The authors would like to thank the reviewer for carefully checking our manuscript and for his valuable comments that allowed to improve the manuscript. 

Yours sincerely,

Vitalijs Radenkovs